# Configuration of Multifunctional Polyimide/Graphene/Fe_3_O_4_ Hybrid Aerogel-Based Phase-Change Composite Films for Electromagnetic and Infrared Bi-Stealth

**DOI:** 10.3390/nano11113038

**Published:** 2021-11-12

**Authors:** Tao Shi, Zhiheng Zheng, Huan Liu, Dezhen Wu, Xiaodong Wang

**Affiliations:** State Key Laboratory of Organic–Inorganic Composites, Beijing University of Chemical Technology, Beijing 100029, China; st956598022@163.com (T.S.); zhengzhihengchn@163.com (Z.Z.); wdz@mail.buct.edu.cn (D.W.)

**Keywords:** electromagnetic and infrared bi-stealth, phase change material, polyimide-based aerogels, graphene/Fe_3_O_4_ hybrid, thermal regulation, microwave absorption

## Abstract

Electromagnetic (EM) and infrared (IR) stealth play an important role in the development of military technology and the defense industry. This study focused on developing a new type of multifunctional composite film based on polyimide (PI)/graphene/Fe_3_O_4_ hybrid aerogel and polyethylene glycol (PEG) as a phase change material (PCM) for EM and IR bi-stealth applications. The composite films were successfully fabricated by constructing a series of PI-based hybrid aerogels containing different contents of graphene nanosheets and Fe_3_O_4_ nanoparticles through prepolymerizaton, film casting, freeze-drying, and thermal imidization, followed by loading molten PEG through vacuum impregnation. The construction of PI/graphene/Fe_3_O_4_ hybrid aerogel films provides a robust, flexible, and microwave-absorption-functionalized support material for PEG. The resultant multifunctional composite films not only exhibit high microwave absorption effectiveness across a broad frequency range, but also show a good ability to implement thermal management and temperature regulation under a high latent-heat capacity of over 158 J/g. Most of all, the multifunctional composite films present a wideband absorption capability at 7.0–16.5 GHz and a minimum reflection loss of −38.5 dB. This results in excellent EM and IR bi-stealth performance through the effective wideband microwave absorption of graphene/Fe_3_O_4_ component and the thermal buffer of PEG. This study offers a new strategy for the design and development of high-performance and lightweight EM–IR bi-stealth materials to meet the requirement of stealth and camouflage applications in military equipment and defense engineering.

## 1. Introduction

Stealth technologies are a combination of technologies that attempt to make military equipment and vehicles, mostly aircraft, less visible or ideally invisible to radar, infrared, sonar, and other detection methods, leading to a practical form of active camouflage [1]. Stealth technology was initially developed for aircraft by using transparent materials or painting with light colors to reduce “visibility” before RADAR was invented. With the invention of RADAR during World War II, stealth became both more necessary and more feasible because aircraft could be effectively detected by RADAR [2]. In recent decades, the development of stealth technology through employing every available method to avoid detection by visible, RADAR, infrared (IR), and acoustic means has attracted tremendous attention in the fields of military science and defense technology, especially for new weapons and military equipment [3]. RADAR and IR detections are the two mainstream means in military reconnaissance. RADAR is the use of reflected electromagnetic (EM) waves in the microwave part of the spectrum to detect targets or map landscapes, whereas IR sensors or thermal imaging cameras can only detect and visualize the targets that have IR radiance characteristics different from their background [4]. In most cases, these two detection technologies are used simultaneously in military practice, and therefore, there is an emergent requirement for modern military systems to develop compatible stealth and integrated technologies that can implement both EM and IR stealth [5,6]. Nevertheless, it is evidently difficult to integrate IR and EM stealth into one system owing to their opposite working principles. Realizing stealth effectiveness requires the integration of many techniques and materials, in which both EM and IR stealth are involved [7]. EM stealth requires that a target such as an aircraft absorbs incident EM wave pulses, actively cancels them by emitting inverse waveforms, deflects them away from receiving antennas, or all of the above. Absorption and deflection are the most important prerequisites of EM stealth [8]. On the other hand, IR radiation is emitted by all matter above absolute zero. Hot materials such as the engine exhaust gases of an aircraft or its wing surfaces heated by friction with the air normally can emit more IR radiation than cooler materials [9]. IR sensors or thermal imaging cameras can only detect and visualize the targets that have IR radiance characteristics different from their background. Meanwhile, the heat-seeking missiles zero in on the IR glow of hot aircraft parts. IR stealth requires that the hot target be kept as cool as possible [10]. Reducing IR emissive level is considered as an effective solution for a target to obtain IR stealth effectiveness. In the case of the design of stealth materials, EM stealth materials are bound to exhibit low reflectivity and high absorption, whereas IR stealth materials should be capable of depressing their thermal emissivity or reducing their surface temperature, or both [11].

A literature survey indicates that there have been many efforts so far made to develop multifunctional materials synchronously with EM and IR stealth capabilities. The traditional method to realize EM and IR bi-stealth is mainly to adopt the microwave absorbers coated by metallic particles with high reflectivity in IR frequencies. Chen et al. [12] reported an investigation on the IR emissivity and EM wave absorption performance of Sm_0.5_Sr_0.5_CoO_3_ perovskites decorated with carbon nanotubes (CNTs) and found that the resultant composites exhibited a remarkable enhanced microwave absorption property along with low IR emissivity compared to pure perovskites. Chu et al. [13] fabricated a type of silver-particle-modified CNT paper/glass-fiber-reinforced polymeric composites for high temperature IR camouflage together with broad EM stealth. Zhang et al. [14] designed a flexible and transparent EM–IR bi-stealth structure by the configuration of a flexible and transparent IR shielding layer combined with a top-layer indium tin oxide as a broadband microwave absorber, realizing low emissivity in IR band and high transparency in EM band accordingly. Xu et al. [15] constructed a novel hybrid metasurface for IR-EM stealth-compatible materials. In this hybrid metasurface, two specially designed metasurface layers that can control the IR emission and microwave absorption were combined to realize EM and IR bi-stealth. Gu et al. [16] developed a type of multifunctional hybrid aerogel based on the melamine skeleton and metal–organic frameworks for IR stealth, thermal insulation, and microwave absorption. This type of hybrid aerogel was found to exhibit strong EM absorbing performance together with superior heat insulating and thermal stealth features. Although these studies made great contributions to the design and development of multifunctional composite materials for EM-IR stealth-compatible applications, they only provided a strategy to realize the EM-IR bi-stealth through a combination of thermal insulation materials and microwave absorbers. However, most of the thermal insulation materials used in these studies lack functionality due to their low electrical and thermal conductivity, indicating a limited IR stealth capability without an effective real time thermal buffer when required.

In this study, we designed and fabricated a novel type of multifunctional composite film based on the polyimide (PI)/graphene/Fe_3_O_4_ hybrid aerogel and polyethylene glycol (PEG) as a phase change material (PCM) for EM and IR bi-stealth. Phase change materials (PCMs) are a class of latent-heat materials that can offer an order of magnitude increase in heat capacity with very small or negligible temperature change compared to traditional sensible heat-storage materials [17,18]. They have attracted a great deal of interest in the IR stealth and thermal camouflage applications due to their unique ability to tune IR emission through passive thermal management [19]. Based on the theory that the total radiated heat energy per unit surface area of a target is directly proportional to the fourth power of its temperature across all wavelengths, IR stealth can be realized by depressing the thermal emissivity of a target or reducing its surface temperature or both [20]. PCMs are evidently able to tune the thermal emissivity of the target through latent heat absorption and controllable release during their phase transitions at varying background temperatures, realizing desirable IR stealth effectiveness [21]. Therefore, it is believed that the utilization of PCMs can generate better IR stealth effectiveness compared to the aforementioned IR stealth technologies. On the other hand, it has been broadly reported that the combination of graphene nanosheets and Fe_3_O_4_ nanoparticles is able to generate prominent microwave-absorption effectiveness because both of them contribute to the attenuation of EM waves through microwave absorption and reflection loss [22]. Multiporous PI aerogel is reconsidered to be a type of promising thermal insulation material with excellent heat resistance, good mechanical performance, high continuous porosity, ultralow density, good flexibility, good chemical stability, and low thermal conductance [23]. Integrating graphene nanosheets and Fe_3_O_4_ nanoparticles into the PI aerogel not only can provide an ultralight, flexible and robust support material for PEG but also can impart excellent microwave absorption and thermal insulating effectiveness to the resultant multifunctional films. It is expected that an innovative integration of PI/graphene/Fe_3_O_4_ hybrid aerogel and PCM can endow the resultant composite material with superior microwave absorption and thermal buffering effectiveness. Furthermore, the EM and IR stealth effectiveness can be further enhanced through flexible and rational configuration of PI/graphene/Fe_3_O_4_ hybrid aerogel films and PI/graphene/Fe_3_O_4_ aerogel/PEG composite films into a multilayer film, extending the application range of this type of multifunctional composite films in practice accordingly. Herein, the fabrication method and formation mechanism of the multifunctional composite films are described in detail, and their microstructure and comprehensive performance are investigated extensively. With an enhanced IR-EM bi-stealth capability, the multifunctional composite films developed in this work are expected to have potential applications for military aircraft, helicopters, surface ships, ground weapons, and other military equipment. This study not only opens a door for the utilization of PCMs in the IR stealth, but also provides a new strategy for the design and development of ultralight, flexible, robust, and high-performance EM-IR bi-stealth functional materials to meet the requirement for the high-efficient EM-IR bi-stealth-compatible applications.

## 2. Materials and Methods

### 2.1. Materials

Pyromellitic dianhydride (PMDA), 4,4′-diaminodiphenyl ether (ODA), *N*,*N*′-dimethylacetamide (DMAc), graphene nanosheets, Ferrous chloride tetrahydrate (FeCl_2_·4H_2_O) and ferric chloride hexahydrate (FeCl_3_·6H_2_O), and triethylamine (TEA) were commercially supplied by Aladdin Reagent Co., Ltd., Shanghai, China. PEG, used as a solid–liquid PCM, was purchased from Tianjin Fuchen Chemical Reagent Co., Ltd., Tianjin, China, and it has a number-average molecular weight of 6000 g/mol and a polydispersity index of 1.79. Ethanol and acetone were purchased from Beijing Chemical Factory Co., Ltd., Beijing, China. All of the chemicals were of analytic grade and used as received without further purification.

### 2.2. Preparation of PI/Graphene/Fe_3_O_4_ Hybrid Aerogel Films

Poly(amic acid) ammonium salt (PAAs) as a water-soluble PI precursor was first synthesized through polycondensation of PMDA and ODA, followed by neutralization with TEA. Figure 1 shows the fabrication method and reaction mechanisms. In a typical procedure, ODA (9.2 g) and DMAc (150.0 mL) were mixed in a 250-mL three-necked round-bottom flask at room temperature with stirring for 30 min to form a homogeneous solution. Then, PMDA (10.1 g) was added into the flask to conduct a polycondensation reaction at 0 °C for 3 h under vigorous agitation. A poly(amic acid) (PAA) solution with a concentration of 12 wt% was achieved at the end of the reaction. Afterward, TEA (24.5 mL) was added into the flask and was stirred at room temperature for 5 h to obtain a PAAs solution. Finally, acetone (200.0 mL) was added into the resultant solution, resulting in the precipitation of reactants. The resultant precipitants were filtrated, washed, and dried in a vacuum oven at 65 °C for 24 h to obtain solid PAAs.

The graphene nanosheets loaded with Fe_3_O_4_ nanoparticles (hereafter graphene/Fe_3_O_4_ hybrid) were prepared through in situ hybridization. In a typical procedure, graphene nanosheets (0.19 g) were dispersed in deionized water (100 mL) under ultrasonication. FeCl_2_·4H_2_O (0.16 g) and FeCl_3_·6H_2_O (0.43 g) were added into the graphene dispersion with mechanical agitation at 50 °C for 20 min under a nitrogen atmosphere. Then, an ammonium hydroxide solution (3.0 mL) was added dropwise into the mixture to maintain the basicity of the aqueous solution at pH 11, and the resulting mixture was stirred for 20 min. Following the completion of co-precipitation, the black precipitate in the solution was collected by a magnet and washed with deionized water several times to obtain the graphene/Fe_3_O_4_ hybrid for future use.

Following the formation of PAAs as a water-soluble polyimide precursor and the graphene/Fe_3_O_4_ hybrid as a functional filler, a series of PI/graphene/Fe_3_O_4_ hybrid aerogel films with a graphene content of 5 wt% and different contents of Fe_3_O_4_ nanoparticles at 0, 2, 4, 6, and 8 wt% were prepared through the freeze-drying and thermal imidization processes as depicted in Figure 1. In a typical procedure, 3.0 g of PAAs, 1.0 mL of TEA, and 26.0 mL of deionized water were mixed in a 100-mL three-necked round-bottom flask under mechanical agitation at 30 °C for 2 h to obtain a clear solution. Graphene/Fe_3_O_4_ hybrid (0.35 g, 0.16 g of graphene nanosheets decorated with 0.19 g of Fe_3_O_4_ nanoparticles) was then added into the flask with continuous stirring for 6 h under a nitrogen atmosphere to obtain a homogeneous PAA/graphene/Fe_3_O_4_ dispersion. The resulting dispersion was cast on a glass plate to form a thin film with a thickness of around 0.5 mm and then subjected to freeze-drying in a lyophilizer (Songyuan Huaxing Technology Development Co., Ltd., Beijing, China) at −80 °C for 12 h to obtain a PAA/graphene/Fe_3_O_4_ hybrid aerogel film. The thermal imidization procedure was further conducted in nitrogen to obtain a PI/graphene/Fe_3_O_4_ hybrid aerogel film with graphene and Fe_3_O_4_ loadings of 5 and 6 wt%, respectively, according to a standard two-stage imidization technique reported in the literature [23,24]. The thermal imidzation process is as follows: heating from room temperature to 150 °C with a heat rate of 2 °C/min and holding at this temperature for 1 h, heating to 300 °C with a heat rate of 2.5 °C/min and holding at this temperature for 2 h, and then naturally cooling down to room temperature. A pure PI aerogel film without any filler was also prepared as a control under the identical conditions for comparison.

### 2.3. Preparation of PI/Graphene/Fe_3_O_4_ Aerogel/PEG Composite Films

A series of PI/graphene/Fe_3_O_4_ aerogel/PEG composite films with a graphene content of 5 wt% and different Fe_3_O_4_ contents of 0, 2, 4, 6, and 8 wt% as well as a PI aerogel/PEG composite film as a control was prepared through a vacuum impregnation method according to the fabrication method shown in Figure 1. In a typical procedure, a PI/graphene/Fe_3_O_4_ aerogel film was immersed into molten PEG at 100 °C for 5 h under vacuum. The molten PEG can easily permeate into the aerogel film through the interconnected channels in the aerogel under the vacuum condition due to its extremely low viscosity at a high temperature of 100 °C. The resultant composite film was immediately wiped with a filter paper to erase the excessive PEG on the film surface. A clean and smooth PI/graphene/Fe_3_O_4_ aerogel/PEG composite film with a thickness of 0.7 mm was finally obtained for future use. In addition, the loading amounts of PEG in the composite films were determined to be 88–90 wt% according to the following equation:(1)$$\mathrm{Loading}\;\mathrm{amount}\;(\mathrm{wt}\%)=\frac{w_{\mathrm{composite}}-w_{\mathrm{aerogel}}}{w_{\mathrm{composite}}}\times 100\%$$
where *w*_composite_ and *w*_aerogel_ are the weights of the PI/graphene/Fe_3_O_4_ aerogel/PEG composite film and PI/graphene/Fe_3_O_4_ hybrid aerogel, respectively.

### 2.4. Characterizations and Measurements

Scanning electron microscopy (SEM) was conducted to observe the morphology and microstructure of composite films using a SUPRA™ 55 scanning electron microscope (Carl Zeiss, Jena, Germany) at an acceleration voltage of 20 kV. Energy-dispersive X-ray (EDX) spectroscopy was performed to analyze the surface elemental compositions and distributions of composite films using a INCAX-Act EDX spectrometer (Oxford Instruments, Abingdon, England). Transmission electron microscopy (TEM) was performed to characterize the morphology and microstructure of graphene/Fe_3_O_4_ hybrids using a JEM 2100 high-resolution transmission electron microscope (JEOL Ltd., Tokyo, Japan) at an acceleration voltage of 100 kV. Atomic force microscopy (AFM) was performed to detect the topology of graphene/Fe_3_O_4_ hybrids using a Dimension ICON atomic force microscope (Bruker AG, Berlin, Germany). Raman spectra were recorded on an inVia confocal Raman microscope (Renishaw, Wotton-under-Edge, UK) equipped with an argon ion laser at an excitation wavelength of 514.5 nm. X-ray powder diffraction (XRD) patterns were recorded on a D/max 2500VB2+/PC X-ray diffractometer (Rigaku, Tokyo, Japan) using a Cu–Kα radiation source. X-ray photoelectron (XPS) spectroscopy was performed to carry out the surface elemental and chemical state analysis of composite films using an ESCALAB–250 XPS spectrometer (Thermo Fisher, Waltham, MA, USA) equipped with a focused monochromatized Al–Kα radiation source. Fourier-transform infrared (FTIR) spectra were recorded on a Nicolet iS5 infrared spectrometer (Thermo Scientific, Waltham, MA, USA) in the wavelength range of 400–4000 cm^−1^ at a resolution of 2 cm^−1^. A PPMS-9 vibrating sample magnetometer (Quantum Design, San Diego, CA, USA) was used to analyze the magnetic properties of graphene/Fe_3_O_4_ hybrids under an applied field of −30,000–30,000 Oe. The magnetization saturation, magnetic retentivity and coercivity were derived from the resultant comparative magnetic hysteresis loops. Mercury intrusion porosimetry was carried out to measure the pore size and volume of aerogel films using an AutoPore V 9600 mercury porosimeter (Micromeritics Instrument, Norcross, GA, USA) in the pressure range of 0.5–62354.60 Psia.

Tensile measurements were carried out using a CMT–4101 electronic universal materials testing machine (MTS Systems, Eden Prairie, MN, USA) at a tensile loading rate of 1 mm/min. Differential scanning calorimetry (DSC) was conducted under nitrogen flux using a Q20 differential scanning calorimeter (TA Instruments, New Castle, DE, USA) with a ramp rate of 10 °C/min. Indium was used as a standard to calibrate the deflections of phase-change temperature and enthalpy. Thermogravimetric analysis (TGA) was carried out using a Q50 thermogravimetric analyzer (TA Instruments, New Castle, DE, USA) with a ramp rate of 10 °C/min under a nitrogen atmosphere. A high-temperature heat impact experiment was carried out to evaluate the shape/form stability and leakage prevention performance of composite films with isothermal heating at 120 °C on a high-precision electronic hot plate, and the aspects of specimens were monitored in real-time using a digital camera (Canon PowerShot SX, Tokyo, Japan). IR thermography was adopted to characterize the thermal regulation performance of composite films during the isothermal heating and natural cooling processes using a Testo^™^ 875–1i IR thermal imaging camera (Testo SE & Co. KGaA, Lenzkirch, Germany). Their surface temperatures were recorded as a function of time as determined from the IR thermographic analysis using the Testo^™^ ComSoft Basic software. IR stealth and thermal camouflage behaviors were examined on different high-temperature targets covered with the aerogel/PEG composite films developed in this work using the same IR thermal imaging camera. The electromagnetic measurements were performed using a PNA-N5244A vector network analyzer (Agilent Technologies, Santa Clara, CA, USA) in the frequency range of 2–18 GHz. In a typical procedure, the composite film was tailored into several pieces and then mixed with paraffin at a mass ratio of 2:1. The as-prepared mixture was compressed into a toroidal shape with an outer diameter of 7.0 mm, an inner diameter of 3.0 mm, and a thickness of about 1.5 mm to fit the sample holder in the instrument. A transmission/reflection method was adopted to measure the electromagnetic parameters using the vector network analyzer, and the permittivity and permeability of the sample were determined from the obtained electromagnetic parameters.

## 3. Results and Discussion

### 3.1. Structural Characterizations of Graphene/Fe_3_O_4_ Hybrid

The graphene/Fe_3_O_4_ hybrid as a microwave absorbing filler was obtained from decorating graphene nanosheets with Fe_3_O_4_ nanoparticles using an in situ hybridization method. Figure 2 shows the SEM, AFM and TEM images of pristine graphene nanosheets and the graphene/Fe_3_O_4_ hybrid. Pristine graphene nanosheets show an ultrathin layered structure with crumpled and rippled textures and curled edges. The thickness of graphene nanosheets was determined to be 2.5–3 nm according to the AFM analysis as seen in Figure 2b. This thickness is equivalent to that of 8–10 layers of graphene. In contrast, Fe_3_O_4_ nanoparticles can be clearly distinguished from the surface of graphene nanosheets as observed in Figure 2d–f. These Fe_3_O_4_ nanoparticles are well dispersed on the graphene surface without any agglomerations, indicating that the graphene nanosheets act as a framework and network to support and separate the loaded Fe_3_O_4_ nanoparticles. Figure 3a shows the FTIR spectra of pristine graphene nanosheets and the graphene/Fe_3_O_4_ hybrid. Both of them were found to exhibit typical graphene characteristic bands by showing the C–OH, C–O–C, C=O absorption peaks in their FTIR spectra. It is worth noting that the IR spectrum of graphene/Fe_3_O_4_ hybrid shows a new Fe–O characteristic absorption peak at 584 cm^−1^ compared to that of pristine graphene nanosheets, indicating the presence of Fe_3_O_4_ nanoparticles on the graphene nanosheets [25]. Figure 3b shows the XRD patterns of the graphene/Fe_3_O_4_ hybrid and pristine graphene nanosheets. Only an intensive diffraction peak was observed at 2*θ* = 26.7° assigned to the (002) plane of graphene from the XRD pattern of pristine graphene nanosheets. However, the XRD pattern of graphene/Fe_3_O_4_ hybrid not only shows a diffraction peak for the (002) reflection of graphene but also exhibit a set of diffraction peaks at 2*θ* = 30.4°, 35.8°, 43.2°, 54.1°, 57.5°, and 63.1° assigned to the (220), (311), (400), (422), (511), and (440) reflections of Fe_3_O_4_ nanoparticles, respectively, according to the standard JCPDS No. 19-0629 [26]. As observed from the Raman spectrum of pristine graphene nanosheets in Figure 3c, three Raman peaks were observed at 1352.2, 1584.6, and 2703.5 cm^−1^ corresponding to the D, G, and 2D vibrational modes of graphene nanosheets, respectively. In contrast, the Raman spectrum of the graphene/Fe_3_O_4_ hybrid shows the extra Raman peaks at 218.9, 280.3, 389.1, 590.9, and 480.1 cm^−1^ associated with the A_1g_, E_g_, and T_2g_ vibrational modes of Fe_3_O_4_ nanoparticles in addition to the Raman peaks of graphene nanosheets [27]. These results confirm the successful formation of the graphene/Fe_3_O_4_ hybrid with an ultrathin laminar structure and small lateral dimensions. Figure 3d shows the comparative magnetic hysteresis loops of Fe_3_O_4_ nanoparticles and a graphene/Fe_3_O_4_ hybrid as well as pristine graphene nanosheets as a control. Pristine graphene nanosheets show no magnetic response, whereas Fe_3_O_4_ nanoparticles exhibit a typical magnetic hysteresis behavior with a saturation magnetic moment of 63.9 emu/g under the applied magnetic field. Furthermore, Fe_3_O_4_ nanoparticles present extremely low magnetic retentivity and coercivity according to the magnetization result, indicating a superparamagnetic feature. It is of importance to note that the graphene/Fe_3_O_4_ hybrid shows a similar magnetic hysteresis behavior to the Fe_3_O_4_ nanoparticles, and their saturation magnetic moment decreases to 27.1 emu/g in the presence of diamagnetic graphene nanosheets. Nevertheless, the negligible coercivity and remanent magnetization reveal that the graphene/Fe_3_O_4_ hybrid is superparamagnetic. In addition, the graphene/Fe_3_O_4_ hybrid is easily dispersed in water to form an aqueous suspension through ultrasonication, and it can be attracted onto the beaker wall using a magnet as observed from the insert in Figure 3d. These results suggest that the fabricated graphene/Fe_3_O_4_ hybrid is ready for use in the PI-based hybrid aerogels as a microwave absorbing filler.

### 3.2. Microstructure and Chemical Characterizations of PI/Graphene/Fe_3_O_4_ Hybrid Aerogel Films

A series of PI/graphene/Fe_3_O_4_ hybrid aerogel films with a graphene content of 5 wt% and different Fe_3_O_4_ content were prepared through polycondensation of PAAs as a water-soluble PI precursor, premixing of PAAs with the graphene/Fe_3_O_4_ hybrid, film casting, freeze-drying, and thermal imidization according to the fabrication method shown in Figure 1. The resultant hybrid aerogel films were found to present ultralight, flexible, foldable, and shape-tunable characteristics as observed from the digital photographs in Figure 1. Figure 4a–h shows the SEM images of pure PI aerogel film and the PI/graphene/Fe_3_O_4_ hybrid aerogel films with different loadings of graphene/Fe_3_O_4_ hybrid. As seen in Figure 4a, pure PI aerogel film was found to show a multiporous 3D cellular structure with a uniform cell size and a thin cell wall. The formation of the 3D cellular microstructure is attributed to the intermolecular interaction of the entangled PI chains, which generates a physical crosslinking effect on the polymer system. This can effectively resist the expansive force in the ice crystal growth process as well as the capillary force during ice sublimation to ensure a stable and robust 3D skeleton for the PI-based aerogels [24]. Similarly, the hybrid aerogel films also exhibit an internal 3D cellular structure as seen in Figure 4b–f. There is almost no visible difference in cellular structure and size between the hybrid aerogel films with various loadings of the graphene/Fe_3_O_4_ hybrid. Moreover, it is observed from the close-up SEM images in Figure 4g,h that the hybrid aerogel films exhibit a compact and robust cell wall. This cell wall not only provides strong support for the whole aerogel framework to load a large quantity of PCM but also facilities an effective load transfer and good resistance to cyclic large-strain deformation. Furthermore, the graphene/Fe_3_O_4_ hybrid can be clearly distinguished as small particles from the cell wall as seen in Figure 4g,h, suggesting a homogenous dispersion of the graphene nanosheets.

The elemental distribution and chemical structure of the PI/graphene/Fe_3_O_4_ hybrid aerogel films were characterized with EDX, FTIR, XRD, and XPS using the hybrid aerogel film containing 5 wt% of graphene nanosheets and 6 wt% of Fe_3_O_4_ nanoparticles as a representative. As observed from the EDX spectrum in Figure 4i, the hybrid aerogel film shows five EDX signals related to the C, O, N, and Fe elements. The corresponding elemental mapping images show the homogeneous elemental distributions of these four elements along with the cell wall of the aerogel as seen in Figure 4i. The FTIR spectrum of the hybrid aerogel film exhibits a series of characteristic absorption peaks at 1772, 1716 and 725 cm^−^^1^ (C=O), 1494 cm^−^^1^ (C=C in benzene ring), 1374 cm^−^^1^ (C–N), 1231 cm^−^^1^ (C–O), 827 cm^−^^1^ (C–H), and 585 cm^−^^1^ (Fe–O), corresponding to PI and graphene/Fe_3_O_4_ hybrid as marked in Figure 5a [28]. There is no evidence that the presence of graphene nanosheets and Fe_3_O_4_ nanoparticles affects the chemical structure of the PI matrix as observed from the FTIR spectrum of the hybrid aerogel film in Figure 5a. The XRD analysis indicates that pure PI aerogel has an amorphous nature due to the presence of only a diffuse peak (Figure 5b). However, most of the diffraction peaks for graphene nanosheets and Fe_3_O_4_ nanoparticles were identified in the XRD pattern of the hybrid aerogel film. In addition, the XPS survey spectrum of the hybrid aerogel film in Figure 6a clearly shows a series of peaks regarding C 1s, O 1s, N 1s, Fe 2p, Fe 3s, and Fe 3p. Furthermore, the C–C, C=C (benzene ring), C=O, C–O, C–N, C–O–C, C–N–C, and –CO–NH– bands as well as Fe(II) and Fe (III) ions were clearly identified from the deconvoluted C 1s, O 1s, N 1s, and Fe 2p spectra in Figure 6b–e [29]. These deconvoluted peaks provide strong evidence of the existence of PI matrix and dispersed graphene/Fe_3_O_4_ hybrid for the hybrid aerogel films. These results confirm the successful fabrication of PI/graphene/Fe_3_O_4_ hybrid aerogel films with the desired morphology, microstructure, and chemical structure.

The porosity characteristics of pure PI aerogel film and PI/graphene/Fe_3_O_4_ hybrid aerogel films were investigated by mercury intrusion porosimetry, and the resultant mercury intrusion and pore size distribution curves are shown in Figure 7. The porous parameters of all samples are summarized in Table 1. It can be seen in Figure 7 that both PI aerogel film and the PI/graphene/Fe_3_O_4_ hybrid aerogel film exhibit a fast increase in the total volume of mercury intrusion at a pore size larger than 40 μm, indicating a macroporous feature of the aerogel films. These macropores also show a wide size distribution ranging from 10 to 100 μm. It should be noted that there is a slight decrease in the mean pore diameter and total immersion volume of the hybrid aerogel films with the addition of the graphene/Fe_3_O_4_ hybrid as observed in Table 1. It seems that the pore diameter and total volume of the hybrid aerogel films tend to decrease with an increase in the Fe_3_O_4_ content. This may be due to the nucleation effect of Fe_3_O_4_ nanoparticles in the process of cell formation, resulting in smaller cells [30]. Furthermore, the introduction of Fe_3_O_4_ nanoparticles as well as graphene nanosheets as inorganic fillers improves the viscosity of the PAAs costing solution, restricting the coalescence and growth of cells [31]. However, the porous parameters seem to be only slightly affected by the introduction of the graphene/Fe_3_O_4_ hybrid. In summary, these porosity characteristics maximize the volume capacity of the hybrid aerogel films to load a great quantity of PEG and meanwhile minimize the leakage of PEG in the molten state.

### 3.3. Morphology and Chemical Characterizations of PI/Graphene/Fe_3_O_4_ Aerogel/PEG Composite Films

PI/graphene/Fe_3_O_4_ aerogel/PEG composite films were fabricated using a simple vacuum impregnation method as depicted in Figure 1. Figure 8 shows the SEM images of the obtained PI aerogel/PEG composite film and PI/graphene/Fe_3_O_4_ aerogel/PEG films with a graphene content of 5 wt% and different Fe_3_O_4_ contents. It is of importance to note that all composite films have been fully impregnated with PEG in their cellular framework, and almost no void is observed. The loading amount of PEG in these composite films were determined to be 88–90 wt%, indicating an extremely high loading of PEG in the PI-based aerogel composite system. Such a high loading quantity can be attributed to a huge pore volume and a large specific surface of the aerogel films, generating a capillary force to anchor PEG macromolecules in the pores together with the hydrogen bond from the cell wall [32].

The chemical and crystalline structures of the PI/graphene/Fe_3_O_4_ hybrid aerogel films were characterized with EDX, FTIR, and XRD using the hybrid aerogel film containing 5 wt% of graphene nanosheets and 6 wt% of Fe_3_O_4_ nanoparticles as a representative. Figure 9 shows the resultant EDX spectrum and elemental mapping images, FTIR spectra, and XRD patterns. As observed from the EDX spectrum in Figure 9a, the hybrid aerogel/PEG composite film not only shows the existence of C, O, N, and Fe elements but also presents a significant increase in the C and O elemental fractions compared to the corresponding hybrid aerogel film. The elemental mapping images also demonstrate that the C and O elements are mainly distributed in the space among the aerogel frameworks. The FTIR spectrum of the hybrid aerogel/PEG composite film exhibits all the characteristic absorption bands of the hybrid aerogel film as shown in Figure 9b. Furthermore, a series of new characteristic absorption peaks can be observed at 3474 and 1645 cm^−1^ for the O–H bond, 2884, 1466, 1342, 961 and 844 cm^−1^ for the C–H bond, and 1097 and 1145 cm^−1^ for the C–O–C bond. These IR characteristic bands are derived from the PEG loaded in the hybrid aerogel film [33,34]. In addition, no evidence was found to show a chemical interaction between the hybrid aerogel and PEG as observed from the IR absorption bands (Figure 9c). This suggests that physical adsorption occurs between the hybrid aerogel film and PEG through a capillary effect, surface tension, and hydrogen bonding. It is also observed from the XRD patterns that the hybrid aerogel/PEG composite film exhibits two intensive diffraction peaks at 2*θ* = 19.2° and 23.4° assigned to (120) and (132) planes of PEG (see Figure 9d) [35]. This suggests that the impregnation of PEG into the micro-sized cellular framework seems not to affect its crystalline structure. These characterization results prove the successful formation of the PI/graphene/Fe_3_O_4_ aerogel/PEG composite films according to our design principle.

### 3.4. Mechanical Performance

Good mechanical performance plays a vital role in the practical application of a aerogel film, especially when it is subjected to frequent bending, stretching, or folding to form a curved surface. A uniaxial tensile experiment was conducted to evaluate the tensile properties of the PI/graphene/Fe_3_O_4_ hybrid aerogel films and PI/graphene/Fe_3_O_4_ aerogel/PEG composite films. Figure 10 shows the resultant stress–strain curves and corresponding experimental data. Pure PI aerogel film was found to present a low tensile strength of 0.67 MPa and a low tensile modulus of 13.7 MPa at a fracture strain of 7.21%, suggesting poor mechanical performance. It is worth noting that the introduction of graphene/Fe_3_O_4_ hybrid generates an effective reinforcement on the PI-based aerogel films. It can be seen in Figure 10a,b that the obtained hybrid aerogel films show an increasing trend in tensile strength and modulus with an increase in the Fe_3_O_4_ content. The hybrid aerogel film containing 5 wt% graphene nanosheets and 8 wt% Fe_3_O_4_ nanoparticles exhibits enhanced tensile strength and modulus of 1.51 and 25.5 MPa, respectively. Such a remarkable improvement in tensile properties may be attributed to the nanoscale reinforcement effect of both nanomaterials. In this nanocomposite system, the load can be transferred effectively across the PI aerogel framework under deformations, preventing crack growth in the aerogel system [36]. It is interesting to note that the PI/graphene/Fe_3_O_4_ aerogel/PEG composite films exhibit a further enhancement in tensile strength and modulus compared to the hybrid aerogel films as seen in Figure 10c,d. This can be explained by the fact that the PEG loaded in the PI-based framework facilitates the transfer, dispersion, and dissipation of stress in the matrix under a tensile load, contributing to an enhancement in tensile performance. Similarly to the hybrid aerogel films, the hybrid aerogel/PEG composite films also exhibit an increasing trend in tensile strength and modulus with an increase in the Fe_3_O_4_ content. This indicates that the tensile performance of the hybrid aerogel/PEG composite films relies on their supporting material, i.e., the hybrid aerogel film. A high-strength hybrid aerogel film can provide stronger mechanical support, resulting in better tensile properties for the composite films. As a result, the hybrid aerogel/PEG composite film containing 5 wt% graphene nanosheets and 8 wt% Fe_3_O_4_ nanoparticles achieved tensile strength of 1.82 MPa and a tensile modulus of 34.26 MPa.

### 3.5. Phase-Change Behavior and Thermal Performance

Phase-change behavior plays an important role in heat energy absorption, storage, and release of a PCM at the desired temperature, speed, and quantity for practical use. The phase-change behavior and thermal performance of PI/graphene/Fe_3_O_4_ aerogel/PEG composite films were investigated by dynamic DSC scans, and pure PEG and PI aerogel/PEG composite films were also examined as controls. Figure 11 shows the resultant DSC thermograms and corresponding thermal analysis data. Both pure PEG and the hybrid aerogel/PEG composite films were found to present similar exothermic and endothermic behaviors in their DSC thermograms as seen in Figure 11a,b, in which the exothermic and endothermic peaks are related to the crystallization and melting phase transitions of PEG, respectively. As observed from the thermal analysis data obtained from DSC measurement (Figure 11c,d), pure PEG exhibits a solidification enthalpy (Δ*H*_c_) of 183.6 J/g at a crystallization peak temperature (*T*_c_) of 35.7 °C a fusion enthalpy (Δ*H*_m_) of 188.7 J/g at a melting peak temperature (*T*_m_) of 61.4 °C. These results indicate that PEG possesses a high latent-heat capacity as an organic PCM. It is of importance to note a slight decrease in *T*_c_ but a slight increase in *T*_m_ taking place in the hybrid aerogel/PEG composite films. The decrease of *T*_c_ can be ascribed to the capillary effect generated by the PI-based framework. Under a capillary force, the motion of PEG macromolecular chains is greatly restricted during the crystallization process, resulting in a higher temperature required for the crystallization of PEG [37,38]. It is noteworthy that the *T*_c_’s of the hybrid aerogel/PEG composite films tend to decrease with an increase in the Fe_3_O_4_ content. This may be due to the reduction of cell volume in the hybrid aerogel, resulting in more serious geometric confinement. On the other hand, the *T*_m_’s of the hybrid aerogel/PEG composite films were also found to increase with an increase in the Fe_3_O_4_ content. This can be explained by the fact that the presence of the PI-based framework postpones the heat transfer of the composite system, delaying the melting process of the PEG loaded in the aerogel. Accordingly, a delayed thermal response occurs in the loaded PEG during the heating process, resulting in an increase in *T*_m_ [39]. It is worth noting that compared to pure PEG, the hybrid aerogel/PEG composite films only exhibit a slight loss in Δ*H*_c_ and Δ*H*_m_. This indicates that there is a high mass fraction of PEG loaded in the aerogel system, imparting a high latent-heat capacity of approximately 159 J/g to the hybrid aerogel/PEG composite films. Such a high heat storage capacity is adequate for practical use in IR stealth and thermal camouflage.

The phase-change durability and long-term heat energy storage-release stability of the PI/graphene/Fe_3_O_4_ aerogel/PEG composite films were investigated through a thermal cycling experiment with consecutive heating-cooling cycles between 0 and 100 °C for 500 cycles. DSC measurements were conducted for the samples after every 50 thermal cycles. Figure 12 shows the obtained DSC thermograms, thermal analysis data and structural characterization results. As seen in Figure 12a, the DSC thermograms of the aerogel/PEG composite films obtained at every 50 thermal cycles are in coincidence with each other. All of them exhibit almost the same location and intensity in their endothermic and exothermic peaks from the first to the last cycle. It can be seen in Figure 12b that there is an extremely small fluctuation in both phase-change temperature and enthalpy with a variation of the thermal cycle number. These results reflect a reversible phase-transition capability of the hybrid aerogel/PEG composite films. With such a reliable phase-change capability, they can implement stable and long-term thermal energy storage in a broad temperature range. FTIR, XRD and SEM characterizations were conducted to evaluate the chemical and structural stability of the hybrid aerogel/PEG composite film before and after 500 thermal cycles. It is of importance to note that the hybrid aerogel/PEG composite film exhibits very similar FTIR spectra before and after the thermal cycling experiment with little visible difference in location and intensity of characteristic absorption bands between the two spectra (see Figure 12c). This indicates that there is no change occurring in the chemical structure of the hybrid aerogel/PEG composite film after suffering from 500 thermal cycles. Moreover, the hybrid aerogel/PEG composite film shows almost the same XRD patterns with similar diffraction intensities before and after the thermal cycling experiment as observed in Figure 12d, indicating an excellent crystalline ability of the PEG loaded in the PI-based aerogels after suffering 500 thermal cycles. In addition, the hybrid aerogel/PEG composite film exhibits a similar morphology and microstructure before and after the thermal cycling experiment as observed from the SEM images in Figure 12e. There is also no visible change in its aspect or discernable leakage of the loaded PEG from the aerogel film. These results demonstrate that the PI-based aerogel skeleton can provide adequate mechanical strength and effective protection for the loaded PEG to undertake multicycle heating and cooling in practical use for IR stealth.

### 3.6. Thermal Stability and Heat-Impact Resistance

The thermal stability of the PI/graphene/Fe_3_O_4_ aerogel/PEG composite films were investigated using TGA, and pure PEG and the PI/PEG composite film were also examined as controls. Figure 13a,b show the obtained TGA and derivative thermogravimetry (DTG) thermograms, respectively. Pure PEG was found to exhibit a typical one-step thermal degradation behavior in a narrow temperature range of 326–410 °C. The pyrolysis leads to an almost complete decomposition of PEG chains with little residual char left at the end of the TGA test. In contrast to pure PEG, both the PI/PEG composite film and the hybrid aerogel/PEG composite films undertake a one-step thermal decomposition along with a secondary mass loss at a higher temperature. The major thermal degradation is derived from the thermal decomposition of the PEG loaded in the aerogels, whereas the secondary minor one is attributed to pyrolysis of the PI skeleton at a much higher temperature of approximately 580 °C. The characteristic temperature at the maximum mass-loss rate (*T*_max_) is generally used as an indicator to reflect the thermal stability of the polymeric material [40]. The *T*_max_’s of all samples were determined from the peak temperatures of their DTG curves as shown in Figure 13b. The hybrid aerogel/PEG composite films were found to exhibit a slight enhancement in thermal stability by showing slightly higher *T*_max_’s than pure PEG. It is also noted in Figure 13c that the hybrid aerogel/PEG composite films show a considerable reduction in thermal conductance compared to pure PEG. These results demonstrate that the PI-based skeleton acts as a physical barrier to prevent the thermal decomposition of the PEG loaded in the aerogels, enhancing the thermal stability of the hybrid aerogel/PEG composite films accordingly.

The shape stability and heat-impact resistance of the PI/graphene/Fe_3_O_4_ aerogel/PEG composite films were evaluated through an isothermal heat-impact experiment, with pure PEG and the PI/aerogel/PEG composite film examined as controls. In this experiment, all samples were cut into square chips and then subjected to an isothermal heat impact on the hot plate at 120 °C, and their appearances were monitored using a digital camera during the isothermal heating process. Figure 14 shows the representative photographs recorded in real-time. Pure PEG tends to fuse when heating up to its melting temperature and then rapidly loses its original shape to flow away. This phenomenon is ascribed to a phase transition from solid to liquid occurring in PEG, resulting in an unstable form in the molten state. In contrast to pure PEG, all of the aerogel/PEG composite films were found to maintain their original forms and shapes well across the heating process. It is worth noting that there is no leakage observed from these aerogel/PEG composite films during the isothermal heating process. These phenomena suggest that the PI-based skeleton can provide effective protection against high-temperature heat impact for the PEG loaded in the aerogel, stabilizing its form and shape, and preventing its leakage. As a result, the hybrid aerogel/PEG composite films can retain a durable solid form even at an operation temperature much higher than the melting point of PEG.

### 3.7. IR Stealth and Thermal Camouflage Behaviors

The thermal buffering and temperature-regulating behaviors of the PI/graphene/Fe_3_O_4_ aerogel/PEG composite film, PI aerogel/PEG composite film, and PI/graphene/Fe_3_O_4_ hybrid aerogel film were investigated comparatively using a facile IR thermographic method. These three samples were subjected to heating up to 90 °C using a hot plate as a heat source and then natural cooling down to room temperature. Their surface temperatures as well as the background temperature were monitored using an IR thermal imaging camera during the heating and cooling processes. Figure 15 shows the resultant IR thermographic images recorded in real-time together with the corresponding temperature evolution with time determined from the IR thermographic analysis. It is interesting to note that these three samples present different color evolution with heating and cooling time as observed from the IR thermographic images shown in Figure 15a. The PI/graphene/Fe_3_O_4_ hybrid aerogel film shows a visible color change from dark blue through light purple to yellow during the heating process, representing a rapid rise in surface temperature. An opposite phenomenon can be observed from the hybrid aerogel film in the natural cooling process, reflecting a drastic decrease in surface temperature without a discernible delay. These results can be confirmed by its temperature evolutions with heating and cooling time derived from IR thermographic analysis. Both the PI aerogel/PEG and the hybrid aerogel/PEG composite films were found to exhibit a lower color change than the PI/graphene/Fe_3_O_4_ hybrid aerogel film during the heating and cooling processes, indicating a delayed temperature change with regard to the hybrid aerogel film. The plots of temperature evolution with time clearly show two temperature hysteresis regions appearing in the heating and cooling processes as observed from Figure 15b,c. These two hysteresis regions are consistent with the melting and crystallization temperature ranges of PEG. The presence of such two temperature hysteresis regions is ascribed to the latent-heat absorption and release of the PEG loading in the aerogel system during the heating and cooling processes.

As reported in the literature, the total quantity of heat energy absorbed or released by a material is normally determined by two forms of energy, i.e., sensible heat and latent heat. Sensible heat energy describes the energy exchange between a system and its surroundings, whereas latent heat represents the changes in the internal heat energy of a system [41]. As an organic PCM, PEG is a typical latent heat-storage material through the phase transitions between solid and liquid. Therefore, both sensible heat and latent heat are involved in the total energy absorption and release of the PI aerogel/PEG and PI/graphene/Fe_3_O_4_ aerogel/PEG composite films. In general, the latent heat resulting from a phase transition in the physical states is many times greater than the sensible heat required to raise a solid or liquid to its melting or boiling temperature. Therefore, such a huge amount of latent heat adsorbed or released can lead to a tardy response to background temperature, generating an extra thermal buffering effect on the composite film. However, the total quantity of thermal energy absorbed or released by the PI/graphene/Fe_3_O_4_ hybrid aerogel film is only dominated by sensible heat energy in the absence of PCM, resulting in a continuous process in the temperature evolution without any hysteresis. It is noteworthy that the PI/graphene/Fe_3_O_4_ aerogel/PEG composite film exhibits a slower temperature evolution and more significant temperature hysteresis compared to the PI aerogel/PEG one as seen in Figure 15. Several investigations demonstrate that graphene is able to enhance the internal photothermal conversion in its composite systems owing to its good near IR absorption ability [42]. Therefore, graphene nanosheets can act as an IR absorbing material to effectively promote the IR absorption of the hybrid aerogels, thus enhancing the thermal response and phase transition of the PEG loaded in the hybrid aerogel films upon a heat source. This improves the heat absorption efficiency of the hybrid aerogel/PEG composite films, reducing their surface temperature more effectively. This is very advantageous to IR stealth and thermal camouflage of a heat source against IR detection.

In general, there is an evident temperature difference between the normal environment and high-temperature target. This leads to a significantly different intensity of thermal IR radiation and generates different IR emissivity accordingly [43]. Such a difference in emissivity can be easily detected by an IR thermal imaging camera or IR sensor. The PI-based aerogel/PEG composite films are believed to control the surface temperature of a high-temperature target to synchronize with its backgrounds through a thermal buffering effect derived from the latent-heat absorption of PEG in the melting phase transition process. The IR stealth and thermal camouflage behaviors of the PI/graphene/Fe_3_O_4_ hybrid aerogel film and hybrid aerogel/PEG composite film were investigated through infrared thermographic detection using a hot plate at 90 °C as a simulated high-temperature target. Figure 16a–d shows the IR thermal images of the high-temperature target covered with different configurations of two types of multifunctional films. It can be seen in Figure 16a that with a multiporous structure and low thermal conductance, the PI/graphene/Fe_3_O_4_ hybrid aerogel film can generate a partial IR stealth effect through heat insulation to block IR emission from the high-temperature target. The PI/graphene/Fe_3_O_4_ aerogel/PEG composite film was found to exhibit a better IR stealth effect than the hybrid aerogel film by showing a target color closer to the background as seen in Figure 16b. This indicates that the thermal buffering derived from the loaded PEG can generate more significant IR stealth effectiveness for the aerogel composite film. It is worth noting that there is a significant enhancement in the IR stealth effect when the configuration of an upper PI/graphene/Fe_3_O_4_ aerogel/PEG composite film and a lower PI/graphene/Fe_3_O_4_ hybrid aerogel film is applied to the high-temperature target (see Figure 16c), indicating a synergetic effect of thermal buffering and heat insulation on IR stealth. According to the thermal buffering and heat insulation mechanism shown in Figure 16f, the lower PI/graphene/Fe_3_O_4_ hybrid aerogel film as a multiporous heat-insulating material not only contributes low thermal conduction for the composite system but also depresses the heat transfer caused by IR radiation. On the other hand, the upper PI/graphene/Fe_3_O_4_ aerogel/PEG composite film offers a thermal buffering effect on the composite system through sensible heat absorption at low temperatures and latent heat absorption at high temperatures. As a result, the double-layered film system can generate coordinative thermal buffer and heat insulation to realize IR stealth and thermal camouflage for the targets in a wide temperature range. It is surprising to note in Figure 16d that the IR thermographic images display the color of simulated high-temperature target almost as same as the background when the sandwich structure-multilayered films with the PI/graphene/Fe_3_O_4_ aerogel/PEG film as a middle layer and the PI/graphene/Fe_3_O_4_ hybrid aerogel film as double-side layers are applied. There is little difference in surface temperature between the high-temperature target and background. In addition, it is also observed in Figure 16e that the configuration of the double-layered or multilayered films can lead to slower growth of the radiant temperature than when the film is used individually, thus providing a longer time for the high-temperature target to achieve IR stealth effectiveness. These results suggest that good IR stealth effectiveness can be achieved through flexible and rational configuration of PI/graphene/Fe_3_O_4_ hybrid aerogel film and PI/graphene/Fe_3_O_4_ aerogel/PEG composite film in future practical use.

### 3.8. Microwave Absorption and EM Stealth Behaviors

Microwave absorption plays a critical role in the realization of EM stealth because RADAR is the use of reflected EM waves in the microwave part of the spectrum to detect targets or map landscapes. EM stealth requires that a target-like craft absorbs incident RADAR pulses, actively cancels them by emitting inverse waveforms, deflects them away from receiving antennas, or all of the above. Both absorption and deflection are the most important prerequisites of EM stealth [44]. In the PI-based hybrid aerogel/PEG composite films designed in this study, graphene nanosheets and Fe_3_O_4_ nanoparticles were included synchronously as microwave absorbing fillers, resulting in the composite of magnetic loss and dielectric loss materials. Such a combination not only can generate multiple losses but also can improve the impedance matching characteristics of the hybrid aerogel/PEG composite films. Figure 17 shows the complex permittivity and complex permeability of the PI/graphene/Fe_3_O_4_ hybrid aerogel/PEG composite films as well as the PI aerogel/PEG composite film as a control. According to the microwave absorption mechanism based on the analysis of complex permittivity (*ε*_r_ = *ε*′ − *jε*″) and complex permeability (*μ*_r_ = *μ*′ − *jμ*″), the real parts, *ε*′ and *μ*′, represent the storage capability of electric and magnetic energy, respectively, whereas the imaginary parts, *ε*″ and *μ*″, reflect the dissipation ability of electric and magnetic energy, respectively [45,46]. It can be seen in Figure 17 that the PI aerogel/PEG composite film shows small values in both real part and imaginary part of its complex permeability and complex permeability over the whole frequency region. This indicates extremely low dielectric and magnetic losses due to the absence of graphene nanosheets and Fe_3_O_4_ nanoparticles. In general, the dielectric loss of a material is related to the polarization relaxation and/or conductive loss, and it can be used to identify the EM energy dissipation ability of a material [47]. The PI/graphene hybrid aerogel/PEG composite film shows a fair increase in dielectric and magnetic losses across the whole frequency range compared to the PI aerogel/PEG composite film. There is a peak of dielectric loss tangent observed at approximately 14.5 GHz (Figure 17c), indicating an excellent dielectric loss in the high frequency range. According to the Debye theory, the dielectric loss ability of a material is also strongly associated with the interfacial and dipole-polarizing effects [48]. The low polarization and electrical isolation natures of PI and PEG resulted in almost no microwave absorption. The dipole polarization in the PI/graphene aerogel/PEG composite film is derived from the multiporous structure and multiple interfaces of the aerogel. Owing to the high conductance and large specific area of graphene nanosheets, the presence of graphene nanosheets contributes to the conductive loss and interfacial effect as important sources for dielectric loss [49]. However, the PI/graphene hybrid aerogel/PEG composite film still shows inadequate dielectric and magnetic losses in the absence of Fe_3_O_4_ nanoparticles compared to the PI/graphene/Fe_3_O_4_ hybrid aerogel/PEG composite films as shown in Figure 17.

It is worth noting that the PI/graphene/Fe_3_O_4_ hybrid aerogel/PEG composite films exhibit significantly enhanced real parts and imaginary parts of their complex permeability along with an improvement in the dielectric loss. Their real parts and imaginary parts were found to continuously increase with an increase in the Fe_3_O_4_ content, followed by an increasing improvement in dielectric loss as seen in Figure 17c. On the other hand, their real and imaginary parts of complex permeability also increase continuously with the increase of Fe_3_O_4_ content, resulting in an increase in magnetic loss [50]. Moreover, there is a series of broad resonance peaks at 6–18 GHz observed from the imaginary part of complex permeability as seen in Figure 17e. These results indicate that the integration of graphene nanosheets and Fe_3_O_4_ nanoparticles generates a synergistic enhancement effect on dielectric and magnetic losses. According to the electromagnetic theory, the dielectric and magnetic losses of the hybrid aerogel/PEG composite films may be attributed to natural resonance, Debye dipolar relaxation, and electron polarization relaxation as well as the interfacial polarization resulting from their multiporous structure and multiple interfaces [51]. Figure 18 shows the scheme of microwave absorption mechanism for the PI/graphene/Fe_3_O_4_ hybrid aerogel films. As depicted in Figure 18, microwave attenuation and impedance matching are two major factors contributing to microwave absorption performance. The decoration of graphene nanosheets with Fe_3_O_4_ nanoparticles attenuates microwave energy through both dielectric loss and magnetic loss. The interfaces between the graphene nanosheets and Fe_3_O_4_ nanoparticles also generate multiple scattering and interfacial polarization, consequently increasing the microwave absorption [52]. Therefore, the decoration of graphene nanosheets with Fe_3_O_4_ nanoparticles not only changes the graphene lamellar structure but also acts as polarized centers to facilitate the electromagnetic energy absorption. Furthermore, both the dielectric and the magnetic losses exhibit dependence on frequency, and they decrease with an increase in frequency [53]. This may be due to the orientation and polarization of random dipoles in the composite system under an external EM field [54]. However, the dielectric polarization effect induced by the dipoles becomes weak if rearrangement of the dipoles cannot match the increasing frequency of the EM waves. As a result, the dielectric response shows a decreasing trend with an increase in frequency. Summarily, these EM characterization results demonstrate that the PI/graphene/Fe_3_O_4_ hybrid aerogel/PEG composite films have a good microwave absorption ability to meet the requirement of EM stealth applications.

To further investigate the microwave absorption characteristics of the PI/graphene/Fe_3_O_4_ hybrid aerogel/PEG composite films, the reflection loss (*R*_L_) could be determined according to the transmit-line theory and calculated using the following equations [55].
(2)$$R_{L}(\mathrm{dB})=20\log\left|\frac{Z_{\mathrm{in}}-Z_{0}}{Z_{\mathrm{in}}+Z_{0}}\right|$$
(3)$$Z_{\mathrm{in}}=\sqrt{\frac{\mu_{r}}{\varepsilon_{r}}}\tanh\left[j\left(\frac{2\pi fd}{c}\right)\sqrt{\mu_{r}\varepsilon_{r}}\right]$$
where *Z*_in_ is the input impedance of the composite films, *Z*_0_ is the impedance in free space, *c* is the velocity of light, *f* is the frequency of EM wave, *ε*_r_ and *μ*_r_ are the complex permittivity and complex permeability, respectively, and *d* is the thickness of the composite film. In general, a higher absolute value of *R*_L_ reflects a stronger ability to absorb EM waves. Figure 19 shows the reflection losses obtained from PI aerogel/PEG composite film, PI/graphene aerogel/PEG composite film, and PI/graphene/Fe_3_O_4_ hybrid aerogel/PEG composite film through calculation with Equations (2) and (3). It can be seen in Figure 19a–c that the microwave absorption properties of these three samples are strongly dependent on their thickness, and their reflection loss peaks shift toward a lower frequency with an increase in the film thickness. As seen in Figure 19a, the minimum reflection losses of the PI aerogel/PEG composite film are all higher than −10 dB across the whole test frequency range at different thicknesses, representing an ability to absorb 90% of the microwaves. The PI/graphene aerogel/PEG composite film shows an improvement in minimum reflection loss up to −22.4 dB at a frequency of 10.1 GHz and a thickness of 1.6 mm (see Figure 19b). This indicates the important contribution of graphene nanosheets to the improvement of microwave absorption performance of the PI/graphene aerogel/PEG composite film through increasing its dielectric loss. As for the PI/graphene/Fe_3_O_4_ hybrid aerogel/PEG composite film, its minimum reflection loss was found to reach −38.5 dB at a frequency of 13.5 GHz and a thickness of 2.8 (see Figure 19c). The bandwidth corresponding to the reflection loss below −10 dB reaches 8.5 GHz in the frequency range of 7.0–16.5 GHz at a thickness of 2.0–3.6 mm. Such a bandwidth covers the wide frequency bands from the C to Ku band, indicating excellent microwave absorption properties of the hybrid aerogel/PEG composite film. It is of importance to note that the bandwidth of the hybrid aerogel/PEG composite film covers the whole X band from 8.0 to 12.0 GHz. This result is highly attractive for military RADAR and direct broadcast satellites due to the high-resolution imaging and accurate target recognition capability [56]. In addition, there is an Fe_3_O_4_-content dependency of minimum reflection loss observed from the hybrid aerogel/PEG composite films as seen in Figure 19d. The minimum reflection loss tends to increase with an increase in the Fe_3_O_4_ content and reaches an optimal value at a Fe_3_O_4_ content of 6 wt%. These results suggest that the PI/graphene/Fe_3_O_4_ hybrid aerogel/PEG composite film developed by this study possesses an outstanding microwave-absorption capability in terms of both the minimum reflection loss and absorption bandwidth and can meet the requirement of EM stealth.

## 4. Conclusions

In summary, we have developed a novel type of multifunctional phase-change composite films based on PI/graphene/Fe_3_O_4_ hybrid aerogel and PEG for EM and IR bi-stealth applications. A series of PI/graphene/Fe_3_O_4_ hybrid aerogel films were first constructed through prepolymerizaton, film casting, freeze-drying, and thermal imidization, and then PEG as a PCM was impregnated into the aerogel framework under a vacuum condition. The resultant hybrid aerogel films not only showed ultralight, robust foldable, and flexible characteristics but also achieved a good microwave absorption capability as a result of the introduction of the broad-band EM-absorbing materials, i.e., graphene nanosheets and Fe_3_O_4_ nanoparticles. The hybrid aerogel films also exhibit a high PEG loading of near 90 wt% thanks to their macroporous structure and large pore volume. The resultant hybrid aerogel/PEG composite films gained a good thermal management capability to implement temperature regulation through thermal energy absorption and release under a latent-heat capacity of over 158 J/g. These multifunctional composite films also exhibit high heat-impact resistance and good thermal cycle stability to meet the requirement of long-term use for EM and IR bi-stealth applications. More importantly, the multifunctional composite films present a wideband absorption capability at 7.0–16.5 GHz and a minimum reflection loss of −38.5 dB. This results in excellent EM and IR bi-stealth performance through the effective wideband microwave absorption of graphene/Fe_3_O_4_ component and the thermal buffer of PEG. Through an innovative integration of PI-based functionalized multiporous materials and PCMs, the resultant multifunctional composite films gained a superior ability to implement EM and IR bi-stealth. This study offers a new strategy for the design and development of high-performance and lightweight EM–IR bi-stealth materials to meet the requirement of stealth and camouflage applications in military equipment and defense engineering.

## Figures and Tables

**Figure 1 nanomaterials-11-03038-f001:**
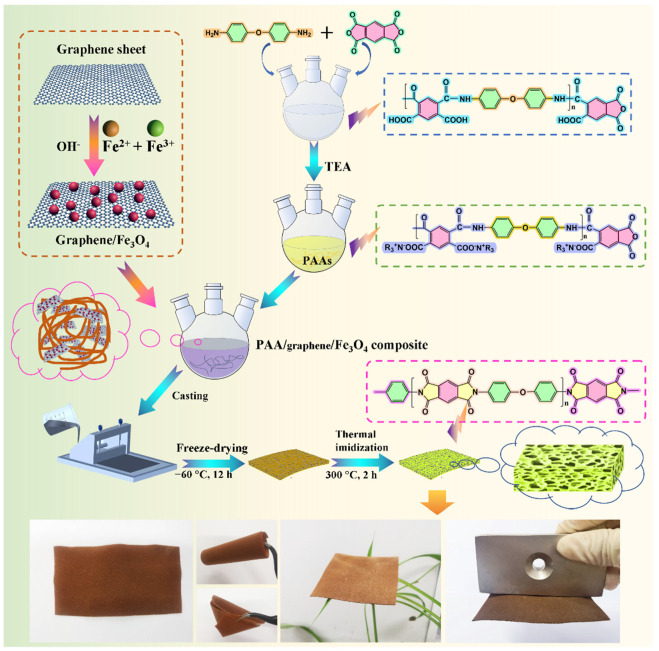
Scheme of fabrication methods and reaction mechanisms of PI/graphene/Fe_3_O_4_ hybrid aerogel film and PI/graphene/Fe_3_O_4_ aerogel/PEG composite film as well as the digital photos of PI/graphene/Fe_3_O_4_ hybrid aerogel film.

**Figure 2 nanomaterials-11-03038-f002:**
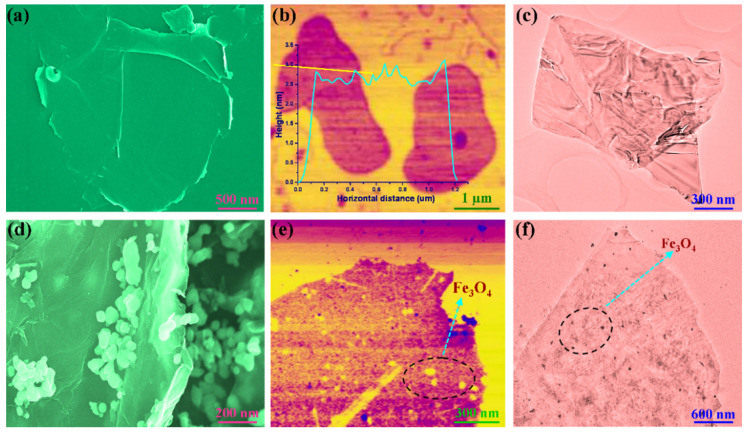
(**a**) SEM, (**b**) AFM, and (**c**) TEM images of pristine graphene nanosheets. (**d**) SEM, (**e**) AFM, and (**f**) TEM images of graphene/Fe_3_O_4_ hybrid.

**Figure 3 nanomaterials-11-03038-f003:**
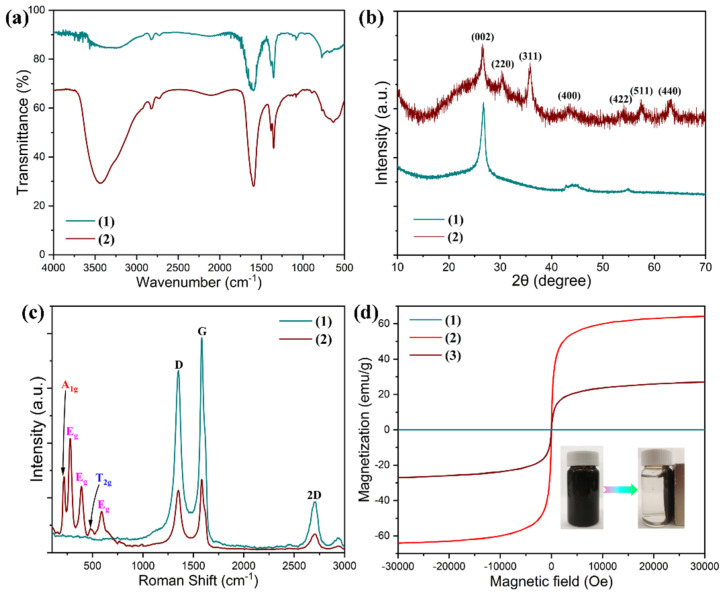
(**a**) FTIR spectra, (**b**) XRD patterns, and (**c**) Raman spectra of (1) pristine graphene nanosheets and (2) graphene/Fe_3_O_4_ hybrid. (**d**) Magnetic hysteresis loops of (1) pristine graphene nanosheets, (2) Fe_3_O_4_ nanoparticles, and (3) graphene/Fe_3_O_4_ hybrid along with the digital photos of the graphene/Fe_3_O_4_ hybrid suspension before and after applied to a magnetic field.

**Figure 4 nanomaterials-11-03038-f004:**
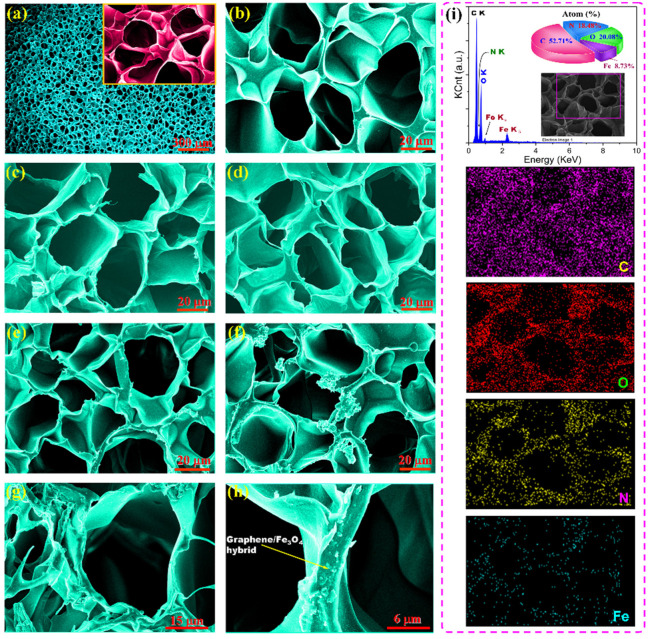
SEM images of (**a**) pure PI aerogel film and the PI/graphene/Fe_3_O_4_ hybrid aerogel films with a graphene content of 5 wt% and different Fe_3_O_4_ contents of (**b**) 0, (**c**) 2, (**d**) 4, (**e**) 6, and (**f**–**h**) 8 wt%. (**i**) EDX spectrum and elemental mapping images of the PI/graphene/Fe_3_O_4_ hybrid aerogel film.

**Figure 5 nanomaterials-11-03038-f005:**
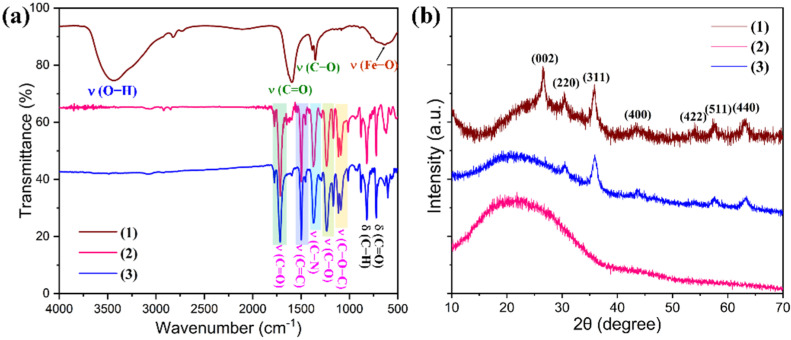
(**a**) FTIR spectra and (**b**) XRD patterns of (1) graphene/Fe_3_O_4_ hybrid, (2) pure PI aerogel film, and (3) the PI/graphene/Fe_3_O_4_ hybrid aerogel film.

**Figure 6 nanomaterials-11-03038-f006:**
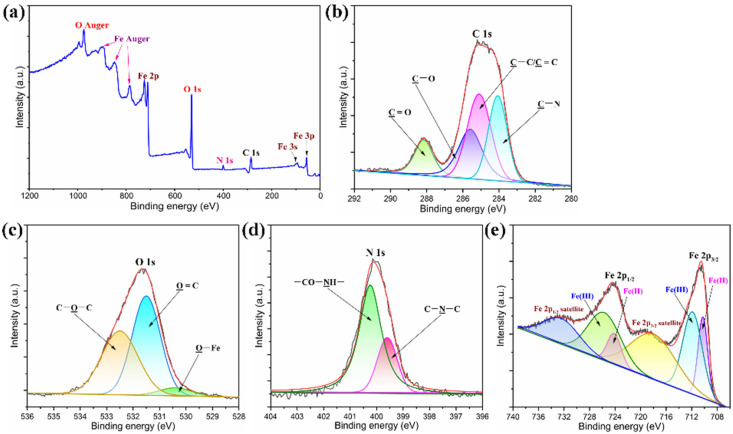
(**a**) XPS survey spectrum of the PI/graphene/Fe_3_O_4_ hybrid aerogel film. High-resolution core-level XPS spectra and deconvoluted curves of the PI/graphene/Fe_3_O_4_ hybrid aerogel film in specific binding energy of (**b**) C 1s, (**c**) O 1s, (**d**) N 1s, and (**e**) Fe 2p.

**Figure 7 nanomaterials-11-03038-f007:**
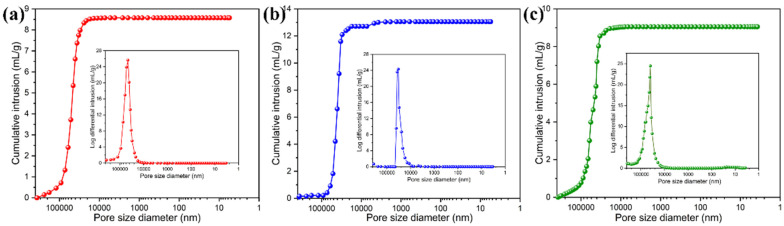
Mercury intrusion and pore size-distribution curves of (**a**) pure PI aerogel film and the PI/graphene/Fe_3_O_4_ hybrid aerogel films with a graphene content of 5 wt% and different Fe_3_O_4_ contents of (**b**) 0 and (**c**) 6 wt%.

**Figure 8 nanomaterials-11-03038-f008:**
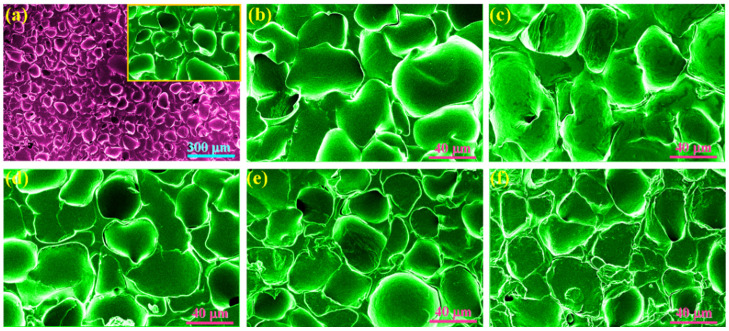
SEM images of (**a**) pure PI aerogel/PEG composite film and the PI/graphene/Fe_3_O_4_ aerogel/PEG composite films with a graphene content of 5 wt% and different Fe_3_O_4_ contents of (**b**) 0, (**c**) 2, (**d**) 4, (**e**) 6, and (**f**) 8 wt%.

**Figure 9 nanomaterials-11-03038-f009:**
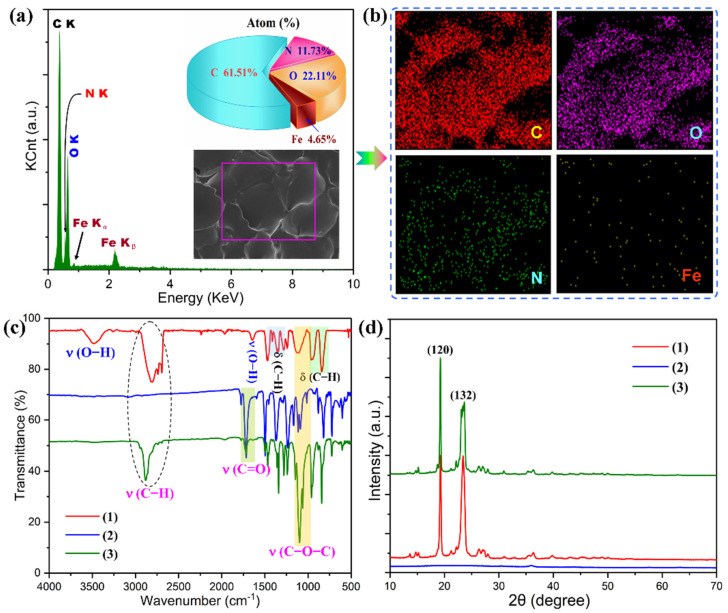
(**a**) EDX spectrum and (**b**) elemental mapping images of the PI/graphene/Fe_3_O_4_ aerogel/PEG composite film. (**c**) FTIR spectra and (**d**) XRD patterns of (1) pure PEG, (2) the PI/graphene/Fe_3_O_4_ aerogel film, and (3) PI/graphene/Fe_3_O_4_ aerogel/PEG composite film.

**Figure 10 nanomaterials-11-03038-f010:**
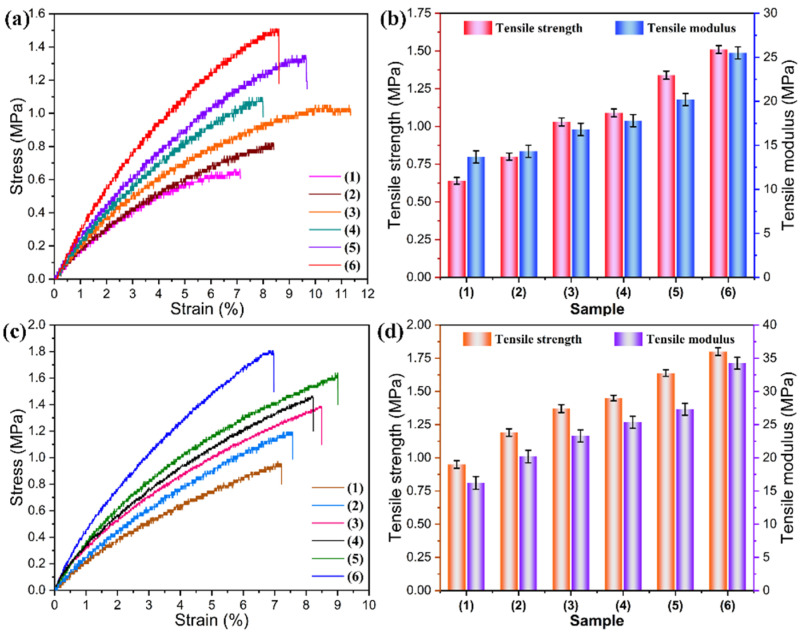
(**a**) Stress–strain curves and (**b**) tensile properties of (1) pure PI aerogel film and PI/graphene/Fe_3_O_4_ hybrid aerogel films with a graphene content of 5 wt% and different Fe_3_O_4_ contents of (2) 0, (3) 2, (4) 4, (5) 6, and (6) 8 wt%. (**c**) Stress–strain curves and (**d**) tensile properties of (1) PI aerogel/PEG composite film and PI/graphene/Fe_3_O_4_ aerogel/PEG composite films with a graphene content of 5 wt% and different Fe_3_O_4_ contents of (2) 0, (3) 2, (4) 4, (5) 6, and (6) 8 wt%.

**Figure 11 nanomaterials-11-03038-f011:**
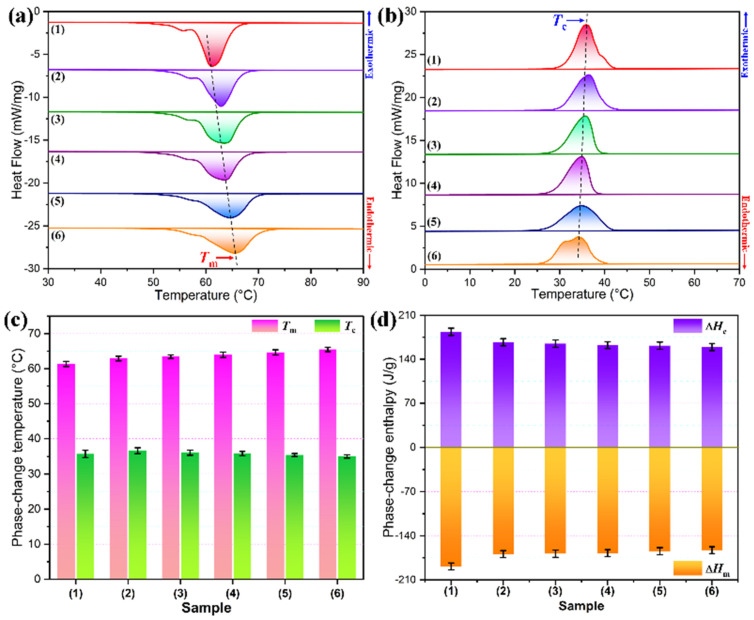
(**a**) DSC cooling thermograms, (**b**) DSC melting thermograms, (**c**) phase-change temperatures, and (**d**) phase-change enthalpies of (1) pure PEG and PI/graphene/Fe_3_O_4_ aerogel/PEG composite films with a graphene content of 5 wt% and different Fe_3_O_4_ contents of (2) 0, (3) 2, (4) 4, (5) 6, and (6) 8 wt%.

**Figure 12 nanomaterials-11-03038-f012:**
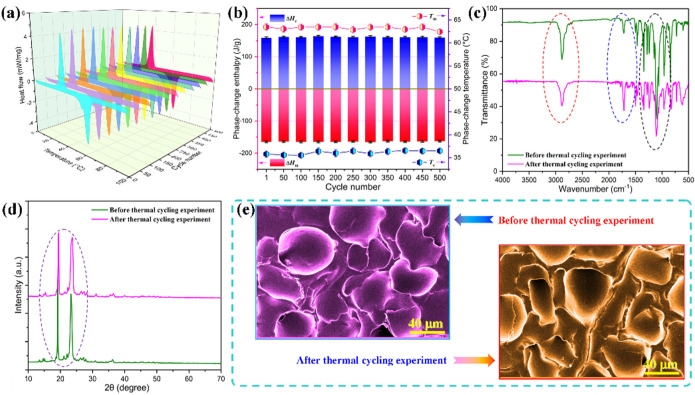
(**a**) Multicycle DSC thermograms and (**b**) phase-change temperatures and enthalpies of the PI/graphene/Fe_3_O_4_ aerogel/PEG composite film containing 5 wt% graphene nanosheets and 6 wt% Fe_3_O_4_ nanoparticles obtained from the thermal cycling experiment at every 50 thermal cycles. (**c**) FTIR spectra, (**d**) XRD patterns, and (**e**) SEM micrographs of the aerogel/PEG composite film before and after the thermal cycling experiment.

**Figure 13 nanomaterials-11-03038-f013:**
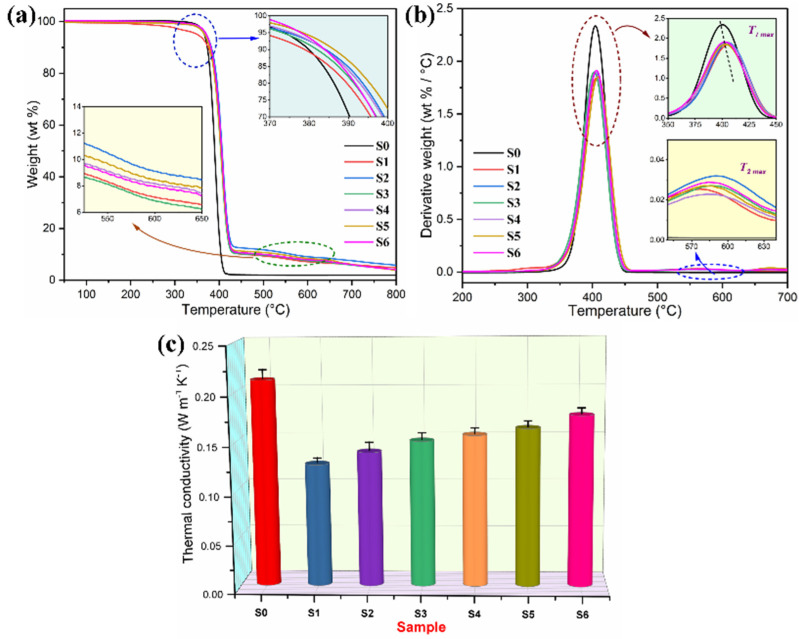
(**a**) TGA thermograms, (**b**) DTG curves, (**c**) Thermal conductivities of (S0) pure PEG, (S1) PI aerogel/PEG composite film, and PI/graphene/Fe_3_O_4_ aerogel/PEG composite films containing 5 wt% graphene nanosheets and (S2) 0, (S3) 2, (S4) 4, (S5) 6 and (S6) 8 wt% Fe_3_O_4_ nanosheets.

**Figure 14 nanomaterials-11-03038-f014:**
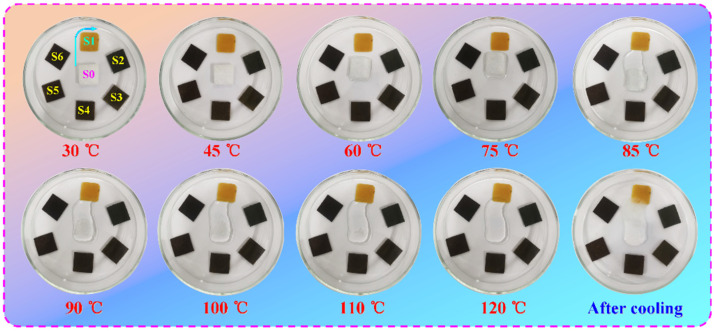
Digital photographs of (S0) pure PEG, (S1) PI aerogel/PEG composite film, and PI/graphene/Fe_3_O_4_ aerogel/PEG composite films containing 5 wt% graphene nanosheets and (S2) 0, (S3) 2, (S4) 4, (S5) 6 and (S6) 8 wt% Fe_3_O_4_ nanosheets recorded during the isothermal heating process.

**Figure 15 nanomaterials-11-03038-f015:**
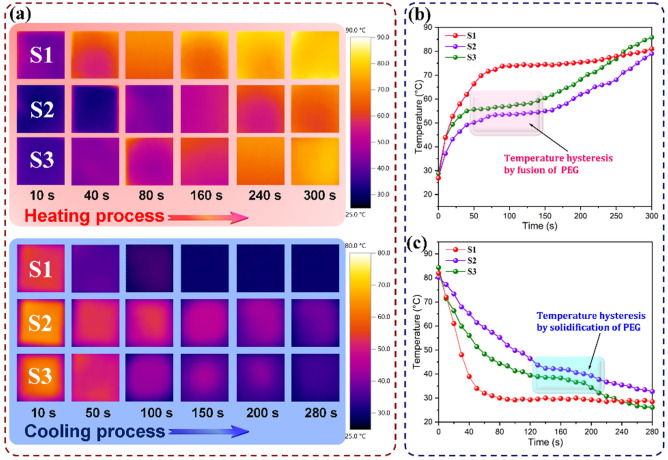
(**a**) IR thermographic images of (S1) PI/graphene/Fe_3_O_4_ hybrid aerogel film, (S2) PI aerogel/PEG composite film, and (S3) PI/graphene/Fe_3_O_4_ aerogel/PEG composite film recorded during the heating and cooling processes. The plots of temperature evolution as a function of time in (**b**) the heating and (**c**) cooling processes determined from IR thermographic analysis formatting.

**Figure 16 nanomaterials-11-03038-f016:**
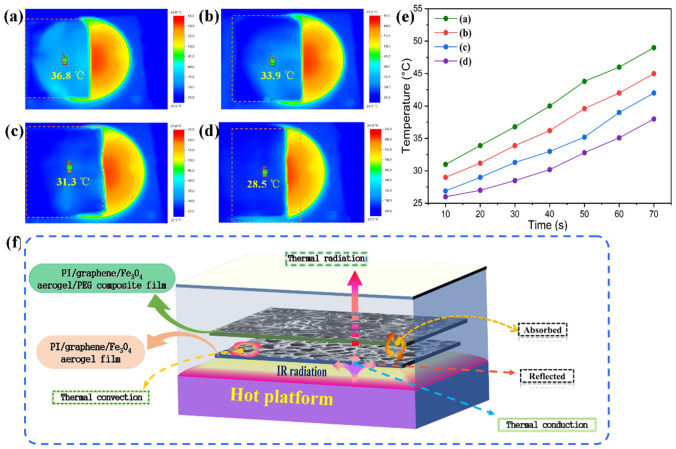
Infrared thermographic images of high-temperature targets partially covered with (**a**) a PI/graphene/Fe_3_O_4_ hybrid aerogel film, (**b**) a PI/graphene/Fe_3_O_4_ aerogel/PEG composite film, (**c**) a lower PI/graphene/Fe_3_O_4_ hybrid aerogel film and an upper PI/graphene/Fe_3_O_4_ aerogel/PEG composite, and (**d**) a PI/graphene/Fe_3_O_4_ aerogel/PEG composite film in the middle and two PI/graphene/Fe_3_O_4_ hybrid aerogel films on the top and bottom. (**e**) The corresponding plots of surface temperature evolution as a function of time obtained from the infrared thermographic analysis. (**f**) Scheme of IR stealth mechanism for the double-layered configuration.

**Figure 17 nanomaterials-11-03038-f017:**
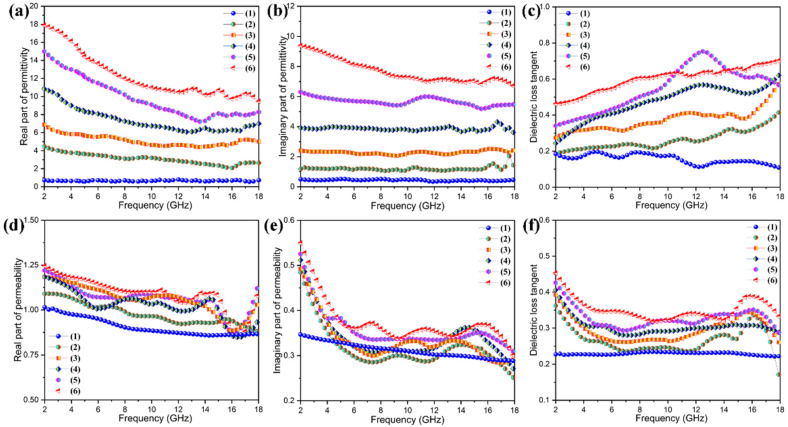
(**a**) Real part, (**b**) imaginary part, and (**c**) dielectric loss tangent of complex permittivity of (1) PI aerogel/PEG composite film and PI/graphene/Fe_3_O_4_ hybrid aerogel films containing 5 wt% graphene nanosheets and (2) 0, (3) 2, (4) 4, (5) 6, and (6) 8 wt% Fe_3_O_4_ nanoparticles. (**d**) Real part, (**e**) imaginary part, and (**f**) dielectric loss tangent of complex permeability of (1) PI aerogel/PEG composite film and PI/graphene/Fe_3_O_4_ hybrid aerogel films containing 5 wt% graphene nanosheets and (2) 0, (3) 2, (4) 4, (5) 6, and (6) 8 wt% Fe_3_O_4_ nanoparticles.

**Figure 18 nanomaterials-11-03038-f018:**
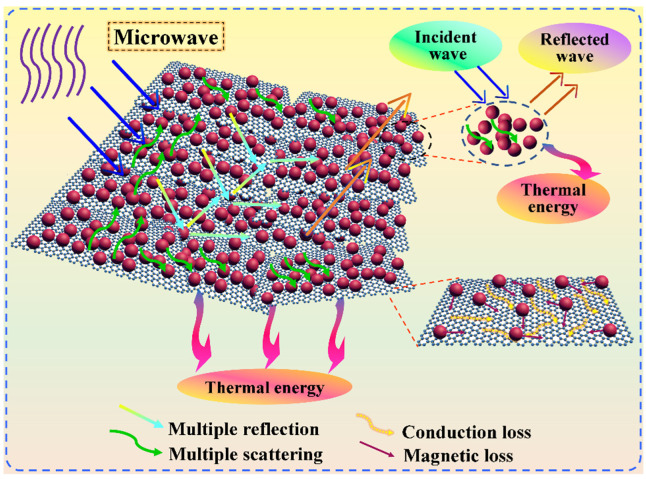
Scheme of microwave absorption mechanism for the PI/graphene/Fe_3_O_4_ hybrid aerogel films.

**Figure 19 nanomaterials-11-03038-f019:**
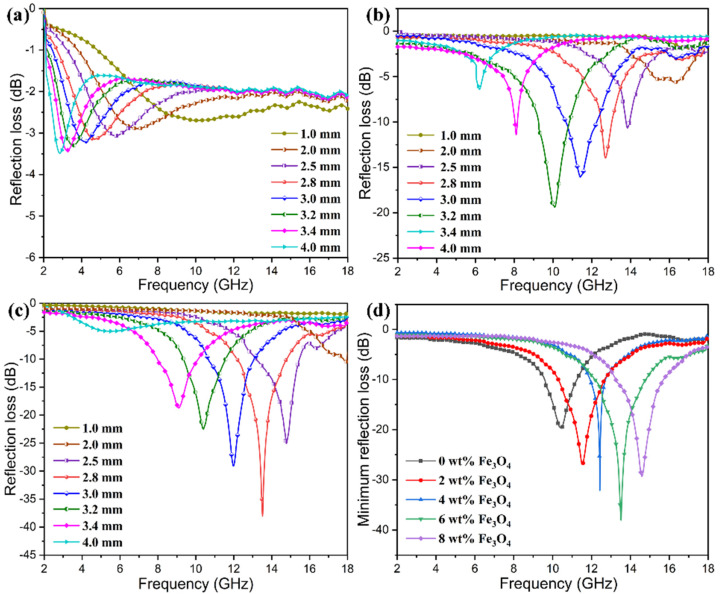
Plots of reflection loss as a function of frequency for (**a**) the PI aerogel/PEG composite film, (**b**) the PI/graphene aerogel/PEG composite film containing 5 wt% graphene nanosheets, and (**c**) PI/graphene/Fe_3_O_4_ hybrid aerogel/PEG composite films containing 5 wt% graphene nanosheets and 6 wt% Fe_3_O_4_ nanoparticles at different film thicknesses. (**d**) Plots of minimum reflection loss as a function of frequency for the PI/graphene/Fe_3_O_4_ hybrid aerogel/PEG composite films containing 5 wt% graphene nanosheets at different contents of Fe_3_O_4_ nanoparticles.

**Table 1 nanomaterials-11-03038-t001:** Porous parameters of PI/Graphene/Fe_3_O_4_ hybrid aerogel films with different loadings of Fe_3_O_4_ nanoparticles.

PI/Graphene/Fe_3_O_4_ Hybrid Aerogel Films	Average Pore Diameter (μm)	Specific Surface Area (m^2^/g)	Total Immersion Volume (mL/g)	Porosity (%)
Pure PI	36.37	1.14	8.77	89.24
PI/graphene/Fe_3_O_4_ (0 wt%)	34.21	1.28	12.92	90.45
PI/graphene/Fe_3_O_4_ (2 wt%)	33.48	1.71	9.64	88.71
PI/graphene/Fe_3_O_4_ (4 wt%)	33.05	2.42	10.81	90.40
PI/graphene/Fe_3_O_4_ (6 wt%)	31.82	3.24	9.44	89.48
PI/graphene/Fe_3_O_4_ (8 wt%)	29.02	5.43	8.11	91.59

## Data Availability

The raw/processed data required to reproduce these findings cannot be shared at this time as the data also forms part of an ongoing study.

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
