# Peer review of "Configuration of Multifunctional Polyimide/Graphene/Fe3O4 Hybrid Aerogel-Based Phase-Change Composite Films for Electromagnetic and Infrared Bi-Stealth"

_nanomaterials, 2021, doi:10.3390/nano11113038_

Round 1
Reviewer 1 Report
The paper “Configuration of Multifunctional Polyimide/Graphene/Fe3O4 Hybrid Aerogel-Based Phase-Change Composite Films for Electromagnetic and Infrared Bi-Stealth” describes an interesting method to prepare multifunctional composite films. The main objective of the study is to investigate the structure and mechanical properties of the obtained composite for Electromagnetic and Infrared Bi-Stealth. It was demonstrated that the new strategy presented in the study can be used to design and develop materials for stealth and camouflage applications. Moreover, the paper is well-written, well-organized and shows very interesting results.
Introduction: The narrative flow was cohesively constructed with the build-up on subject concerning the research gap of previous studies about the recent challenges on producing the desired properties of multifunctional materials for EM and thermal camouflage
Methods The experimental section was clearly described with all the important details and information which is good for replicability of other researchers.
Results and Discussions The results were well-presented showing different characterization methods in order to investigate the relationship between the structure and properties of the developed material.
Based on the above and the general presentation of the paper I recommend the paper for publication to the Nanomaterials.
Author Response
Dear Reviewer 1:
Thank you very much for your kind comments on our manuscript. According to your suggestion, there is no further revision needed.
Sincerely yours
Xiaodong Wang
A professor of materials science and engineering
Beijing University of Chemical Technology
Reviewer 2 Report
General comments Though manuscript is lengthy, but it lacks substance. Typical run of the mill paper on EMI shielding. Manuscript needs major revisions. Specific comments 1. Introduction is long but pretty weak. In lines 66 to 87, authors listed previous studies (unnecessarily) but never justified why this study is necessary and how their work is different or better than previous reports. 2. Authors used a highly inappropriate term like 'aerogel' in this manuscript but their PI/graphene/Fe3O4 hybrid is nothing but a foam. It is like calling commercially available melamine foam as 'melamine aerogel' or PU foam mattress as "PU aerogel" mattress. 3. Thirdly, there are serious shortcomings in experiments. Example: Authors never measured the number of layers in graphene. What is the state of dispersion of Graphene/Fe3O4 in PI? Studies on degree and state of dispersion must be included in revised manuscript. Authors say "heating at 150°C for 1 h, heating at 300 °C for 2 h, and then cooling down to room temperature". What is the heating rate? Why this two step procedure. Justify. Doesn't prolonged heating at 300C for 2h degrade PI? 4. The section "2.3. Preparation of PI/graphene/Fe3O4 aerogel/PEG composite films" is confusing. Authors say "was immersed into molten PEG at 100 °C for 5 h in a vacuum oven". Isn't PEG a viscous liquid? What was molecular weight distribution of PEG used. How much PEG was impregnated into PI/graphene/Fe3O4 aerogel? What was the effect of graphene/Fe3O4 on weight gain? 5. What was thickness of PI/graphene/Fe3O4 hybrid aerogel film? More details about EMI testing must be included in revised manuscript. There are many more problems with this manuscript, but above are enough to recommend major revision.
Author Response
Dear Reviewer 2:
Thank you very much for your critical comments and valuable suggestions on our submission. These suggestions are very helpful for revising and improving our manuscript. We have paid careful attention to these suggests and have modified the manuscript thoroughly according to your suggestions. We have presented all the changes highlighted in the revised manuscript and explained these revisions point by point as follows:
(1) Thank you very much for your valuable suggestion. We have refined our manuscript especially for the section that you pointed out. We have also provided a comparison between the previous reports and our work according to your suggestion.
(2) Thank you very much for your good suggestion. We have modified the item “aerogel” and replaced it with the new item “foam” in the revised manuscript according to your suggestion.
(3) Thank you very much for your valuable suggestions on these issues. We have provided more discussions about the dispersion state of graphene in the revised manuscript according to your suggestion. Hereby, we would like to explain about the study on the dispersion state of graphene in this paper. We have attempted to evaluate the dispersion state of graphene using TEM but found that it was difficult to get an ultrathin slice from the foam for TEM observations. In this case, we have to characterize the dispersion state of graphene with SEM. The close–up SEM images of the cell wall in Figure 4h clearly indicated the homogeneous dispersion of graphene in the hybrid foams. Moreover, we have provided detailed descriptions for the processing procedure and technology of the hybrid foam films in the revised manuscript according to your suggestion. We would also like to explain that the two-step imidization processing for the fabrication of the PI-based foam films used in this paper is a standard processing and technology of PI imidization that have widely reported in the relevant literature. The PI materials have extremely high heat resistance with an upper work temperature of over 400 °C. Therefore, they are completely able to undertake an imidization temperature of 300 °C over a long time.
(4) Thank you very much for your valuable suggestion on this issue. We have provided more information about the preparation of PI/graphene/Fe3O4 aerogel/PEG composite films in the “2.3. Section” of the revised manuscript according to your suggestion.
(5) Thank you very much for your valuable suggestion on this issue. We have described the thickness of the foam films and also provided more information about the EMI testing in the revised manuscript according to your suggestion.
Sincerely yours
Xiaodong Wang
A professor of materials science and engineering
Beijing University of Chemical Technology
Reviewer 3 Report
This study focused on developing a new type of multifunctional composite films based on polyimide (PI)/graphene/Fe3O4 hybrid aerogel and polyethylene glycol (PEG) as a phase change material (PCM) for EM and IR bi-stealth applications. The composite films were successfully fabricated by constructing a series of PI-based hybrid aerogels containing different content of graphene nanosheets and Fe3O4 nanoparticles through prepolymerizaton, film casting, freeze-drying, and thermal imidization, followed by loading molten PEG through vacuum impregnation.
The article can be accepted with minor changes.
Please increase figures clarity.
Conclusion is not obvious. Please add some insights on how your research adds to state of the art. Please outline your contribution to state of the art.
Applications may be substantiated here.
Any prospectus opened by this research?
You can add some references from nanomaterials as:
https://doi.org/10.3390/nano10061168
https://doi.org/10.3390/nano11010086
https://doi.org/10.3390/nano11081871
https://doi.org/10.3390/nano11102612
Author Response
Dear Reviewer 3:
Thank you very much for your kind comments and helpful suggestions on our paper. We have made a careful revision on our manuscript according to your suggestions and presented all the changes highlighted in the revised manuscript. We also have explained the revisions point by point as follows:
(1) Thank you very much for your valuable suggestion. We would like to explain that we have already provided all of the figure images with a high resolution. It is speculated that the scant images appearing in our manuscript might result from imaging compression during a conversion of the file from a word version to the PDF version in the submission system. We believe that all of the figure images will be very clear in the formal publication version.
(2) Thank you very much for your valuable suggestion. We have provided some insights on how our research adds to state of the art for the Conclusion section in the revised manuscript according to your suggestion.
(3) Thank you very much for your good suggestion. We have provided the applications substantiated for the composites developed by our study in the revised manuscript according to your suggestion.
(4) Thank you very much for your good suggestion. We have provided the prospectus opened by this study in the revised manuscript according to your suggestion.
(5) Thank you very much for your good suggestion. We have quoted the references that you recommended in the revised manuscript according to your suggestion.
Round 2
Reviewer 2 Report
General comments
Manuscript is not suitable for Nanomaterials. Major revisions.
Specific comments
In reply to my comments autjors replied that " In this case, we have to characterize the dispersion state of graphene with SEM. The close–up SEM images of the cell wall in Figure 4h clearly indicated the homogeneous dispersion of graphene in the hybrid foams." But Fig. 4h is run of the mill SEM image of a foam which raises the question of how did authors distinguished between polymer and graphene?
Fig. 4(i) shows EDX spectrum and elemental mapping images of the PI/graphene/Fe3O4, but these maps are of which sample from "different Fe3O4 contents of (b) 0, (c) 2, (d) 4, (e) 6, and (f–h) 8 wt %.".
Secondly it is surprising to see no apparant change in cell structure, cell density, cell size with varying Fe3O4 concentration. "Figure 4. SEM images of (a) pure PI aerogelfoam film and the PI/graphene/Fe3O4 hybrid aerogel foam films with a graphene content of 5 wt % and different Fe3O4 contents of (b) 0, (c) 2, (d) 4, (e) 6, and (f–h) 8 wt %."
Thirdly there are serious questions regarding "Figure 8. SEM images of (a) pure PI aerogelfoam/PEG composite film and the PI/graphene/Fe3O4 aerogelfoam/PEG composite films with a graphene content of 5 wt % and different Fe3O4 contents of (b) 0, (c) 2, (d) 4, (e) 6, and (f) 8 wt %". First how did authors managed to take SEM of PEG liquid filled PI foam. And did authors measure the percentage of PEG in "PI/graphene/Fe3O4 aerogelfoam/PEG composite films". Authors must also explain how the morphology changes from 'open cell' to 'closed cell' after PEG impregnation.
Though in most instances authors have used 'foam' but there are still some instances where the inappropriate term "aerogel" is still used in the manuscript. Authors must carefully revise their manuscript.
There are many more problems with this manuscript but above are enough to recommend major revisions again.
Author Response
Dear Reviewer 2:
Thank you very much for your critical comments and valuable suggestions on our submission. These suggestions are very helpful for revising and improving our manuscript. We have paid careful attention to these suggests and have modified the manuscript thoroughly according to your suggestions. We have presented all the changes highlighted in the revised manuscript and explained these revisions point by point as follows:
(1) Thank you very much for your good question in this issue. We have marked the graphene/Fe3O4 hybrid in the cell wall of the form in Figure 4h, where the Fe3O4-decorated graphene nanosheets can be clearly distinguished.
(2) Thank you very much for your concern in this issue. We have specialized the test sample of PI/graphene/Fe3O4 hybrid foam films for EDX FTIR XRD measurements by the statement “the PI/graphene/Fe3O4 hybrid foam films were characterized with EDX, FTIR, XRD, and XPS using the hybrid foam film containing 5 wt % of graphene nanosheets and 6 wt % of Fe3O4 nanoparticles as a representative” in line 342–345 in the revised manuscript.
(3) Thank you very much for your good questions on this issue. The changes in the cell structure, cell density, cell size of the composite films cannot be visibly distinguished from the SEM images in Figure 4 but can be detected by mercury intrusion porosimetry with the resultant data listed in Table 1. In this table, you may find the changes in the pore parameters of the composite films.
(4) Thank you very much for your good questions on this issue. Please allow us to explain the fact in our study. The hybrid foams were impregnated with the molten PEG at a high temperature, and then the obtained composite films were cooled down to room temperature. At this moment, the impregnated PEG was frozen into the solid. Therefore, the SEM images were obtained from the solidified PEG in the foams. The weight percentages of PEG in the forms have been measured and reported in lines 202–203 and 403–405 in the revised manuscript. Moreover, the morphology did not change from 'open cell' to 'closed cell' after PEG impregnation as seen in the SEM images in Figure 8. The PEG is solid in the cells, so it looks like closed cells. In fact, the foam system still has open cells.
(5) Thank you very much for your concern in this issue. We have checked over the whole text and find that there is no item “aerogel” in the text. However, there are several items “aerogel” in the title of cited papers in the reference section. These items cannot be changed because they are original items in the title of the references.